# Learning Juntas under Markov Random Fields

**Gautam Chandrasekaran**
gautamc@cs.utexas.edu
UT Austin

**Adam R. Klivans**
klivans@utexas.edu
UT Austin

## Abstract

We give an algorithm for learning $O(\log n)$ juntas in polynomial-time with respect to Markov Random Fields (MRFs) in a smoothed analysis framework where only the external field has been randomly perturbed. This is a broad generalization[1] of the work of Kalai and Teng, who gave an algorithm that succeeded with respect to smoothed *product* distributions (i.e., MRFs whose dependency graph has no edges). Our algorithm has two phases: (1) an unsupervised structure learning phase and (2) a greedy supervised learning algorithm. This is the first example where algorithms for learning the structure of undirected graphical models have downstream applications to supervised learning.

## 1   Introduction

A function $f : \{0,1\}^n \to \{0,1\}$ is a $k$-junta if it depends only on $k$ of the $n$ input coordinates. The junta learning problem, introduced by Blum and Langley [Blu94, BL97] in 1994, is as follows: given random samples labeled by an unknown $k$-junta, output a classifier that closely approximates the $k$-junta. The problem of learning $k$-juntas is one of the most well-studied problems in computational learning theory over the last three decades. It is considered a notoriously difficult challenge to learn juntas with runtime and sample complexity $n^{o(k)}$ and has important applications in pseudorandomness and cryptography (e.g., [ABW10])

The problem of learning juntas highlights the difficulty of designing learning algorithms that can succeed in the presence of a large number of irrelevant features (i.e., the $n - k$ irrelevant coordinates). Most prior work has focused on learning juntas when the marginal distribution is uniform over the hypercube. Observe that a brute force search over all subsets of $k$ variables is possible in time $O(\binom{n}{k})$. In the search for faster algorithms, [MOS04, Val12] gave algorithms that run in time approximately $n^{0.7k}, n^{0.6k}$ respectively when given uniform random examples. There is evidence to suggest that a runtime of $n^{\Omega(k)}$ is unavoidable, as there is a lower bound of $\binom{n}{k}$ in the statistical query (SQ) framework of Kearns [Kea93] and also cryptographic lower bounds [ABW10]. It is a major open problem to find an algorithm for this problem with run time and sample complexity $n^{o(k)}$.

### 1.1   Beyond the Worst Case: Smooth Product Distributions

To bypass the above hardness results, learning juntas has been studied in the smoothed analysis framework of Spielman and Teng [ST04]. In particular, Kalai and Teng [KT08] introduced the notion of a $(c, \sigma)$-smooth product distribution, which is a product distribution with mean vector of the form $\mu = \bar{\mu} + \Delta$ where $\bar{\mu} \in [-c, c]^n$ is adversarially chosen and $\Delta$ is randomly sampled from the uniform distribution on $[-\sigma, \sigma]^n$. Under these smoothed distributions (with high probability over the smoothing), they showed that it is possible to learn depth $k$ decision trees in time $\text{poly}(2^k, n)$[2].

---

[1] the generalization is in the distributional assumption, Kalai and Teng's result also gives a polynomial time learning algorithm for log-depth decision trees.

[2] A similar statement for juntas was also observed in [MOS04], see Fact 15

The main takeaway from their result is that if the marginal distribution is a slightly perturbed version of the uniform distribution, the task of junta learning becomes easy. This suggests that the lower bound for learning juntas is extremely brittle and that juntas are efficiently learnable over most product distributions.

A major drawback of all the aforementioned results is that they require the marginal distribution to be product. It is not clear how realistic this assumption is, as most real world data has interdependencies between variables. In [KST09], the authors asked if the smoothed analysis paradigm could be extended beyond product distributions.

We answer this question positively, and our main contribution is an efficient algorithm for learning $O(\log n)$ juntas with respect to Markov Random Fields (MRFs) with $O(\log n)$-degree dependency graphs and smoothed external fields. This is a broad generalization of the Kalai et al. result, as (smoothed) product distributions correspond to trivial MRFs where the underlying dependency graph has no edges.

## 1.2 Beyond Products: Markov Random Fields

The class of distributions we study are undirected graphical models, also known as Markov Random Fields. These models– most famously the Ising model– have played a central role in probabilistic modeling and statistical physics.

**Definition 1.1** (Undirected Graphical Model). *An undirected graphical model $D$ with dependency graph $G$ is a probability distribution over $\{0,1\}^n$ such that for $X \sim D$, $X_i$ is conditionally independent of the remaining coordinates of $X$ when the conditioning is on $\{X_j \mid (i,j)$ is an edge in $G\}$.*

By the Hammersley-Clifford Theorem [CH71], every distribution satisfying the above property (with the additional assumption that the density is positive everywhere) has a density function of the following form:

$$\Pr_{X \sim D}[X = x] \propto \exp\left(\sum_{S \in C(G)} \psi_S(x)\right) = \exp\left(\psi(x)\right) \tag{1}$$

where $t \in [n]$, $C(G)$ is the set of cliques of $G$ and $\psi_S$ are functions that only depend on the coordinates of $x$ in $S$. Any distribution of the above form is called a Markov Random Field. The factorization $\psi$ of $D$ is a polynomial of degree at most $d$ where $d$ is the degree of $G$. The famous Ising model corresponds to the case when the degree of $\psi$ is two. The linear part of the polynomial $\psi$ is called the *external field*.

As mentioned above, these distributions strictly generalize product distributions. A product distribution is a graphical model where the graph contains only isolated vertices. The uniform distribution corresponds to an MRF with factorization $\psi = 0$. An arbitrary product distribution corresponds to an MRF with a linear function as the factorization.

Note that again, a brute force algorithm running in time $O(\binom{n}{k})$ exists for learning juntas over MRF distributions. The question we investigate is if runtimes of the form $\text{poly}(2^k, n)$ are possible for perturbed versions of these distributions. We show that this is indeed the case for the following notion of $(\lambda, \sigma)$-smooth MRFs where the external field of an adversarially chosen MRF is perturbed.

**Definition 1.2** (($\sigma, \lambda$)-smooth MRF). *Let $\lambda \in \mathbb{R}$ and $\sigma \in (0, 1/2)$. A Markov random field $D$ is a $(\sigma, \lambda)$-smooth MRF if the factorization polynomial of $D$, denoted by $\psi$ is of the form*

$$\psi(x) := \bar{\psi}(x) + \sum_{i=1}^n \Delta_i x_i$$

*where $\Delta_i = \log(1 + \alpha_i)$ for iid $\alpha_i \sim \text{Unif}([-\sigma, \sigma])$ and $\left\|\partial_i \bar{\psi}\right\|_1 \leq \lambda$ for all $i \in [n]$.*

In the above definition, $\bar{\psi}$ is the factorization of the adversarially chosen MRF and the upper bound on the norm of its derivatives is a standard assumption. This can be interpreted as a multiplicative perturbation of the density function as we have that

$$\Pr_{X \sim D}[X = x] \propto \Pr_{X \sim \bar{D}}[X = x] \prod_{i \in [n]} (1 + \alpha_i x_i)$$

where $\bar{D}$ is the MRF with factorization $\bar{\psi}$. We note that *we only perturb the external field* (as compared to perturbing all coefficients) of the adversarially chosen factorization polynomial in this model. We show that this mild perturbation is sufficient for efficient learnability. We also note that the bound $\left\|\partial_i \bar{\psi}\right\|_1 \leq \lambda$ is a natural non-degeneracy condition and is a generalization of the condition of $p$-biasedness (see Claim 2.4) which prior work [KT08, KST09, BDM20] assume for smooth product distributions. Markov random fields with such a bound on derivatives are sometimes referred to as *bounded width* models and are exactly the class of MRFs for which efficient structure learning (dependency graph recovery) results are possible [Bre15, KM17, HKM17, VMLC16, SW12].

## 1.3 Our Results

We now state our theorem on junta learning over smooth Markov random fields.

**Theorem 1.3.** *Let $\mathcal{D}$ be a labelled distribution such that the marginal distribution is a $(\lambda, \sigma)$-smooth MRF with a known dependency graph $G$ of degree at most $d$ and the labelling function is a $k$-junta. Then, Algorithm 2 run with $N = \Omega(\text{poly}(\log n, \exp(\lambda(d + k)), 2^{d+k}, \sigma^{-k}, 1/\delta))$ samples from $\mathcal{D}$, graph $G$ and appropriately chosen threshold will run in time at most $\text{poly}(n, N)$ and learn the junta exactly, with probability at least $1 - \delta$ over the samples and smoothing of $\mathcal{D}$. In particular, for $d, k \leq O(\log n)$ and $\lambda, \sigma = O(1)$, the algorithm runs in polynomial time.*

This addresses an open question raised in [KST09] about extending their smoothed analysis framework beyond product distributions.

**Remark 1.4.** *In the setting where the dependency graph $G$ is not known, one can first recover $G$ using existing structure learners for bounded-degree MRFs [KM17, HKM17] and then apply our algorithm. The sample complexity and run-time of these algorithms are $\text{poly}(n^t, 2^\lambda)$, where $t$ is the degree of the factorization polynomial of the MRF. This preprocessing is required only once and can be reused across multiple supervised learning tasks with respect to the same distribution. We note that to the best of our knowledge, this is the first example where algorithms for structure learning graphical models have downstream applications to supervised learning.*

**Remark 1.5.** *Our result is also tolerant to random classification noise as the underlying algorithm falls in the Statistical Query (SQ) framework [Kea93].*

## 1.4 Related Work

**Learning and Testing Juntas** The problem of learning juntas was introduced by Blum and Langley [Blu94, BL97] in 1994. The first non-trivial algorithm for learning over the uniform distribution was given by Mossel, O'Donnell, and Servedio [MOS04] who improved the naive $\binom{n}{k}$ runtime from exhaustive search to roughly $n^{0.7k}$. This run-time was improved by Valiant [Val12] to approximately $n^{0.6k}$. The $n^{\Omega(k)}$ runtime is widely believed to be optimal for uniform distribution learning, as there are statistical query and cryptographic lower bounds [Kea93, ABW10]. Another well studied problem related to juntas is that of junta testing. Here, the goal is to identify if an input function is a junta, or far from one, where the algorithm is given query access. The first algorithm was given by Parnas, Ron and Samorodnitsky [PRS01] where they give an algorithm for dictator (1-junta) testing. The first algorithm for $k$-junta testing was given by Fischer et al. [FKR+04] and later improved to almost optimal query complexity by Blais [Bla08, Bla09].

**Smoothed Analysis and Learning** The study of smoothed analysis of algorithms was initiated by Spielman and Teng [ST04] to theoretically study the empirical success of the Simplex algorithm which has exponential worst case run time. Their framework has subsequently been applied to various settings to analyze the good average case performance of various algorithms which have intractable worst case performance. Applying this framework to learning theory, Kalai and Teng [KT08] gave a polynomial time algorithm for PAC learning $O(\log n)$-depth decision trees over smoothed product distributions. Kalai, Samorodnitsky and Teng [KST09] extended the idea to polynomial time algorithms for PAC learning DNFs and agnostically learning decision trees with random examples. Brutzkus, Daniely and Malach [BDM20] proved that the empirically successful ID3 algorithm efficiently learns juntas over these distributions. More recent work applying the framework of smoothed analysis to learning theory include [HRS20, HRS22, CKK+24].

**Learning from Random Walk Examples**   A complementary beyond worst case model for supervised learning is that of learning from random walk samples. This was introduced by Bshouty et al. [BMOS03] where they give a polynomial time algorithm for PAC learning DNFs when given correlated samples corresponding to consecutive steps of an appropriate random walk over the cube whose stationary distribution is uniform. An algorithm for agnostic $O(\log n)$-juntas in polynomial time over the same model was given by Jackson and Wimmer [JW09]. This framework was further generalized to MRFs by Kanade and Mossel [KM15] where they give a polynomial time algorithm for learning $O(\log n)$-juntas but require correlated samples from a rapidly mixing Gibbs random walk whose stationary distribution is an MRF. We note that our algorithm requires only i.i.d. samples from the underlying MRF.

**Learning Markov Random Fields**   Starting with the work of Chow and Liu [CL68] on learning tree Ising models, the problem of learning graphical models has been studied extensively [WLR06, AKN06, BMS08, NBSS10, TR14]. Bresler [Bre15] obtained the first efficient structure learning algorithm for Ising models over bounded degree graphs, although with suboptimal sample complexity. This algorithm was generalized to higher order MRFs by Hamilton, Koehler and Moitra [HKM17]. Vuffray et al. [VMLC16] gave the first algorithm for learning Ising models with near-optimal sample complexity but with suboptimal runtime. Klivans and Meka [KM17] gave the first algorithm that achieves both near-optimal runtime and sample complexity. Other recent related works on structure learning MRFs in various settings include [WSD19, GKK19, PSBR20, MMS21, DKSS21, DDDK21, BGPV21, GMM24, CK24].

## 2   Preliminaries

**Definition 2.1** ($k$-junta). *A function $f : \{0,1\}^n \to 0,1$ is said to be a $k$-junta if there exists a function $g : \{0,1\}^k \to \{0,1\}$ and a set $S \subseteq [n]$ with $|S| = k$ such that $f(x) := g(x_S)$.*

Given a graph $G$, we use $N_G(i)$ to denote the neighbours of $i$ in $G$. Given a distribution $D$ with random variable $x \sim D$, we use $\mathbb{E}_{x \sim D}[.]$ to denote expectations over these variables. We drop the distribution and random variable from the subscript when it is clear from context. Similarly, given a set $S$, we use $\mathbb{E}_S[.]$ to denote the expectation over the uniform distribution on the set $S$.

A useful property that we require is that of $\delta$-unbiasedness.

**Definition 2.2** (Unbiased distributions). *Let $\delta \in [0,1]$. A distribution $D$ is said to be $\delta$-unbiased if for all $b \in \{0,1\}$, $i \in [n]$ and $x \in \{0,1\}^{n-1}$, it holds that*

$$\Pr_{X \sim D}[X_i = b \mid X_{-i} = x] \geq \delta$$

**Fact 2.3.** *Let $D$ be an MRF with factorization polynomial $\psi$. Then, for $i \in [n]$ and $x \in \{0,1\}^{n-1}$, it holds that $\Pr_{X \sim D}[X_i = 1 \mid X_{-i} = x] = \sigma(\partial_i \psi(x))$.*

It is easy to see that the MRFs we consider in this paper are sufficiently unbiased (proof in Appendix A).

**Claim 2.4.** *Let $D$ be a $(\sigma, \lambda)$-smooth MRF for $\lambda \in \mathbb{R}$ and $\sigma \in (0, 1/2)$. Then, it holds that $D$ is $\frac{\exp(-\lambda)}{4}$-unbiased.*

The following is a useful consequence of the unbiasedness property (proof in Appendix A).

**Lemma 2.5.** *Let $\delta \in (0,1)$. Let $D$ be a $\delta$-unbiased distribution. Then, for any sets $S, T \subseteq [n]$ such that $S \cap T = \phi$ and any $y \in \{0,1\}^{|S|}, z \in \{0,1\}^{|t|}$, it holds that $\Pr_{X \sim D}[X_T = t \mid X_S = s] \geq \delta^{|T|}$*

## 3   Algorithm and Analysis

Throughout this section, let $f$ be the ground truth junta that depends on $k$ bits. Let $\mathcal{D}_{\mathbf{x}}$ denote the marginal distribution. The joint distribution over features and labels, denoted by $\mathcal{D}$, is a distribution on $\{0,1\}^n \times \{0,1\}$, where $(x, y)$ drawn from $\mathcal{D}$ is obtained by sampling $x$ from $\mathcal{D}_{\mathbf{x}}$ and setting $y = f(x)$.

## 3.1 Prior Work: Learning over Smooth Product Distributions

Before describing our algorithm for learning juntas over smooth MRFs, we first discuss the prior work on learning juntas over smooth product distributions [KT08, MOS04, BDM20] and explain how their algorithms work. Let $\mathcal{D}_{\mathbf{x}}$ be a product distribution with $\mathbb{E}[x_i] = \mu_i + \Delta_i$ for all $i$ where $\Delta_i$ is uniform in $[-\sigma, \sigma]$. They first define the quantity $I(i)$ as $I(i) := \left| \mathbb{E}_{(x,y)\sim D}[yx_i] - \mathbb{E}_{(x,y)\sim D}[y]\,\mathbb{E}_{x\sim\mathcal{D}_{\mathbf{x}}}[x_i] \right|$ for any index $i \in [n]$. Observe that $f$ can always be uniquely expressed as $f(x) = x_i \cdot g_i(x_{-i}) + h_i(x_{-i})$ where $g_i$ and $h_i$ depend on at most $k-1$ variables. Also, $x_i$ is a relevant variable if and only if $g_i$ is non-zero. They showed that

$$
\begin{aligned}
I(i) &= \left| \mathbb{E}[yx_i] - \mathbb{E}[y]\,\mathbb{E}[x_i] \right| = \left| \mathbb{E}[x_i]\,\mathbb{E}[f(x) \mid x_i = 1] - \mathbb{E}[f(x)]\,\mathbb{E}[x_i] \right| \\
&= \left| \mathbb{E}[x_i] \right| \cdot \left| \mathbb{E}[g_i(x_{-i}) \mid x_1 = 1] \cdot (1 - \mathbb{E}_D[x_i]) + \mathbb{E}[h_i(x_{-i}) \mid x_i = 1] - \mathbb{E}[h_i(x_{-i})] \right| \\
&= \left| \mathbb{E}[x_i](1 - \mathbb{E}[x_i]) \cdot \mathbb{E}[g_i(x_{-i})] \right| = \left| \mathbb{E}[x_i](1 - \mathbb{E}[x_i]) \right| \cdot \left| g_i((\mu + \Delta)_{-i}) \right|. \quad (2)
\end{aligned}
$$

The first three equalities follow from the fact that $f(x) = x_i \cdot g_i(x_{-i}) + h_i(x_{-i})$. The penultimate equality uses the fact that $x_i$ is independent from $x_{-i}$ and hence $\mathbb{E}[h_i(x_{-i}) \mid x_i = 1] = \mathbb{E}[h_i(x_{-i})]$. The last equality follows from treating the function $g_i$ as a polynomial of degree at most $k-1$ and using the product nature of the distribution to conclude that $\mathbb{E}[\prod_{i\in S} x_i] = \prod_{i\in S}(\mu_i + \Delta_i)$. Thus, to lower bound $I(i)$, it suffices to lower bound $g_i((\mu + \Delta)_{-i}))$. Clearly, we have that $I(i) = 0$ for $i$ that is not relevant as $g_i$ is the zero polynomial. For relevant $i$, they used the following lemma from [KT08] on anticoncentration of polynomials to show that $I(i) \geq \delta^2 (\sigma)^{2k}$ with probability at least $1 - \delta$ (for more details, see the proof of Lemma 11 in [BDM20]).

**Lemma 3.1** (Lemma 4 from [KST09]). *Let $c, \sigma \in \mathbb{R}$. Let $p : \mathbb{R}^n \to \mathbb{R}$ be degree $\ell$ multilinear polynomial $p(x) := \sum_{|S|\leq \ell} \hat{p}(S) \prod_{i\in S} x_i$. Suppose there exists a set $S$ with $|S| = \ell$ and $|\hat{p}(S)| \geq c$. Then. for iid $x_i \sim \mathrm{Unif}([-\sigma, \sigma])$, it holds that*

$$
\Pr_{X\sim\mathrm{Unif}([-\sigma,\sigma]^n)}[|p(X)| \leq c\sigma^\ell \cdot \epsilon] \leq 2^\ell \sqrt{\epsilon}.
$$

Since $I(i)$ is sufficiently large for relevant indices, by taking empirical estimates of this statistic from $\mathrm{poly}((2/\sigma)^k)$ samples, one can find the relevant variables. Then, given the set of $k$ relevant variables, even a brute force algorithm (constructing a truth table for all $2^k$ possible combinations) has sample complexity and runtime that only scales with $\mathrm{poly}(n, (2/\sigma)^k)$.

## 3.2 Learning Over Smooth MRFs: Our Techniques

The product structure was crucial in the previous subsection for two main reasons. First, it was used to argue that $\mathbb{E}[g_i(x_{-i})]$ is a polynomial in $\mu + \Delta$. Second, the independence of $x_i$ and $x_{-i}$ was essential in showing that $I(i) = 0$ for irrelevant indices. The former was important to guarantee non trivial correlation for relevant indices, while the latter was essential for distinguishing relevant from irrelevant variables.

We now try to extend this approach beyond products. Henceforth, the marginal distribution $\mathcal{D}_{\mathbf{x}}$ that we consider is a $(\sigma, \lambda)$-smooth MRF (Definition 1.2) with factorization $\psi$. By our definition of smooth MRFs, we will show with additional technical work that the first property ($\mathbb{E}[g_i]$ is a polynomial in perturbations) from earlier is still qualitatively true when we move beyond product distributions. In contrast, the second ($I(i) = 0$ for irrelevant variables) is more fundamentally tied to product structure and does not extend to general distributions. Non-product distributions inherently allow correlations between variables, so an index that is irrelevant to a junta may still exhibit non-trivial correlation with the label. As a result, any algorithm that selects variables purely based on their correlation with the labels risks including irrelevant variables. To address this, we must go beyond the correlation statistic $I(i)$.

Before we can go further, we need some additional notation. A restriction is a string $\rho \in \{0, 1, *\}^n$. Let $\mathrm{supp}(\rho)$ denote the support of the restriction defined as $\mathrm{supp}(\rho) = \{i \in [n] \mid \rho_i \neq *\}$. The size of a restriction $\rho$ denoted by $|\rho|$ is the size of the set $\mathrm{supp}(\rho)$. Given a distribution $D$ on $\{0, 1\}^n$ and a restriction $\rho$, the restricted distribution $\mathcal{D}_\rho$ is obtained by conditioning $\mathcal{D}$ on the event $\{X_i = \rho_i, \text{ for all } i \in \mathrm{supp}(\rho)\}$. Similarly, given a set $S \subseteq \{0, 1\}^n$ and a restriction $\rho \in \{0, 1, *\}$, we use $S_\rho$ to denote the set $S_\rho := \{x \in S \mid x_i = \rho_i, \text{ for all } j \in \mathrm{supp}(\rho)\}$. Given a function

$f : \{0,1\}^n \rightarrow \{0,1\}$ and a restriction $\rho \in \{0,1,*\}^n$, the restricted function $f_\rho$ is defined as $f_\rho(x) := f(x_\rho)$.

To go past the correlation statistic and design an algorithm for learning over MRFs, we use a key structural property of MRFs: the *Markov* property. Recall from Definition 1.1 that the variables $x_i$ and $x_{-i}$ are conditionally independent when conditioned on $x_{j \in N_G(i)}$ where $G$ is the dependency graph of the MRF. This property motivates the statistic of measuring correlation between $x_i$ and $y$ after conditioning by the neighbors of $i$. Formally, for $i \in [n]$ and restricting $\rho \in \{0,1,*\}^n$ with $\mathsf{supp}(\rho) = N_G(i)$, we define the statistic $I(i,\rho)$ as $I(i,\rho) := \left| \mathbb{E}_{\mathcal{D}_\rho}[yx_i] - \mathbb{E}_{\mathcal{D}_\rho}[y]\, \mathbb{E}_{\mathcal{D}_\rho}[x_i] \right|$ where $\mathcal{D}_\rho$ is the joint distribution conditioned on the event that $x_{\mathsf{supp}(\rho)} = \rho_{\mathsf{supp}(\rho)}$. Let $R$ be the set of relevant variables for $f$. We show the following properties for the statistic $I(i,\rho)$:

1. for all $i \notin R$, for all restrictions $\rho$ with $\mathsf{supp}(\rho) = N_G(i)$, it holds that $I(i,\rho) = 0$ (Claim 3.3),

2. for all $i \in R$, with probability $1 - \gamma$ over the smoothing of $\mathcal{D}_{\mathbf{x}}$, there exists a restriction $\rho$ with $\mathsf{supp}(\rho) = N_G(i)$ such that $|I(i,\rho)| \geq \gamma^2 \cdot \left((\sigma \exp(-\lambda)/16)^{k+2}\right)$ (Claim 3.4).

The first claim follows almost immediately from the Markov property. The second claim requires more technical work as the MRF density is quite complicated when compared to the product example sketched in Section 3.1. The proofs of these claims are in Section 3.3. Once we show these claims, the rest of the algorithm is almost immediate: we estimate these statistics empirically for all indices and all restrictions of their neighborhoods and pick the indices for which the statistic is large (Algorithm 1). We note that sampling from these conditional distributions (using straightforward rejection sampling) costs us an $\exp(\lambda d)$ factor in the sample complexity (where $d$ is the max degree of the dependency graph).

---

**Algorithm 1:** FindRelevantVariables$(S, G, \tau)$

---

1 **Input:** Sample set $S \subseteq \{0,1\}^n \times \{0,1\}$, Dependency Graph $G$, Threshold $\tau$
2 $\mathrm{Rel} \leftarrow \phi$
3 **for** $i \in [n]$ **do**
4     **for** $\rho \in \{0,1,*\}^n$ *such that* $\mathsf{supp}(\rho) = N_G(i)$ **do**
5         $I_S(i,\rho) \leftarrow \left| \mathbb{E}_{S_\rho}[yx_i] - \mathbb{E}_{S_\rho}[y]\, \mathbb{E}_{S_\rho}[x_i] \right|$
6         **if** $|I_S(i,\rho)| > \tau$ **then**
7             $\mathrm{Rel} \leftarrow \mathrm{Rel} \cup \{i\}$
8         **end**
9     **end**
10 **end**
11 **Return:** $\mathrm{Rel}$

---

---

**Algorithm 2:** LearnJunta$(S, G, \tau)$

---

1 **Input:** Sample set $S \subseteq \{0,1\}^n \times \{0,1\}$, Dependency Graph $G$, threshold $\tau$
2 $\mathrm{Rel} \leftarrow$ FindRelevantVariables$(S, G, \tau)$
3 Find Empirical Risk Minimizer $\hat{f}$ out of all functions that only depend on $\mathrm{Rel}$
4 **Return:** $\hat{f}$

---

Finally, we run a brute force learner on these indices (Algorithm 2). This is where we use the fact that these distributions are unbiased (Claim 2.4 and Lemma 2.5). More specifically, the brute-force learner requires $O(\exp(\lambda k))$ samples (owing to unbiasedness) to see all possible fixings of the relevant indices. This implies our main theorem (proof in Appendix B). Note that the final hypothesis we output is exact and has zero error with high probability.

**Theorem 3.2.** *Let $\mathcal{D}$ be a labelled distribution over $\{0,1\}^n \times \{0,1\}$ such that $\mathcal{D}_{\mathbf{x}}$ is a $(\lambda, \sigma)$-smooth MRF with dependency graph $G$ of degree at most $d$ and the labelling function is a $k$-junta. Then, Algorithm 2 run with $N = \Omega(\mathrm{poly}(\log n, \exp(\lambda(d+k)), 2^{d+k}, \sigma^{-k}, 1/\delta))$ samples from $\mathcal{D}$, graph $G$ and appropriately chosen threshold $\tau$ will output a hypothesis $\hat{f}$ such that $\mathbb{E}_{(x,y) \sim \mathcal{D}}[\hat{f}(x) \neq y] = 0$ with probability at least $1 - \delta$ over the samples and smoothing of $\mathcal{D}$.*

### 3.3 Proofs

We now prove the two claims (Claim 3.3 and Claim 3.4). Recall that for each $i \in [n]$, we have that $f(x) = x_i \cdot g_i(x_{-i}) + h_i(x_{-i})$. For any $\rho \in \{0, 1, *\}^n$ with $\mathsf{supp}(\rho) = N_G(i)$, we have that

$$\left| \mathop{\mathbb{E}}_{\mathcal{D}_\rho} [y x_i] - \mathop{\mathbb{E}}_{\mathcal{D}_\rho} [y] \mathop{\mathbb{E}}_{\mathcal{D}_\rho} [x_i] \right| = \left| \mathop{\mathbb{E}}_{\mathcal{D}_\rho} [x_i] \mathop{\mathbb{E}}_{\mathcal{D}_\rho} [f(x) \mid x_i = 1] - \mathop{\mathbb{E}}_{\mathcal{D}_\rho} [f(x)] \mathop{\mathbb{E}}_{\mathcal{D}_\rho} [x_i] \right|$$

$$= \left| \mathop{\mathbb{E}}_{\mathcal{D}_\rho} [x_i] \right| \cdot \left| \mathop{\mathbb{E}}_{\mathcal{D}_\rho} [g_i(x_{-i}) \mid x_1 = 1] \cdot (1 - \mathop{\mathbb{E}}_{\mathcal{D}_\rho} [x_i]) + \mathop{\mathbb{E}}_{\mathcal{D}_\rho} [h_i(x_{-i}) \mid x_i = 1] - \mathop{\mathbb{E}}_{\mathcal{D}_\rho} [h_i(x_{-i})] \right|.$$

We now use the Markov property. We have that for $(x, y) \sim \mathcal{D}_\rho$ with $\mathsf{supp}(\rho) = N_G(i)$, it holds that $x_i$ and $x_{-i}$ are independent. Thus, we have that $\mathbb{E}_{\mathcal{D}_\rho}[h_i(x_{-i}) \mid x_i = 1] = \mathbb{E}_{D_\rho}[h_i(x_{-i})]$. Thus, we obtain that

$$|I(i, \rho)| = \left| \mathop{\mathbb{E}}_{\mathcal{D}_\rho} [x_i] \cdot (1 - \mathop{\mathbb{E}}_{\mathcal{D}_\rho} [x_i]) \right| \cdot \left| \mathop{\mathbb{E}}_{\mathcal{D}_\rho} [g_i(x_{-i}) \mid x_1 = 1] \right| = \left| \mathop{\mathbb{E}}_{\mathcal{D}_\rho} [x_i] \cdot (1 - \mathop{\mathbb{E}}_{\mathcal{D}_\rho} [x_i]) \right| \cdot \left| \mathop{\mathbb{E}}_{\mathcal{D}_\rho} [g_i(x_{-i})] \right| \tag{3}$$

Recall that for indices $i$ not relevant to $f$, we have that $g_i = 0$. Thus, we obtain the following claim.

**Claim 3.3.** *Let $i \in [n]$ be such that $f$ does not depend on index $i$. For any restriction $\rho$ with $\mathsf{supp}(\rho) = N_G(i)$, it holds that $I(i, \rho) = 0$.*

We are now ready to prove that for every relevant index $i$, there is a restriction $\rho$ of its neighbours such that $|I(i, \rho)|$ is sufficiently large. The first observation is that it suffices to bound $\mathbb{E}_{D_\rho}[g_i(x_{-i})]$ as the distribution $\mathcal{D}_{\mathbf{x}}$ is unbiased. After this the proof has two main parts. First, we show by writing out the densities and using appropriate algebraic manipulations that this quantity is lower bounded by a polynomial in the smoothing variables. Finally we use polynomial anticoncentration from Lemma 3.1 to complete the proof.

**Claim 3.4.** *Let $i \in [n]$ be such that $f$ depends on index $i$. Then, there exists a restriction $\rho$ with $\mathsf{supp}(\rho) = N_G(i)$ such that $|I(i, \rho)| \geq \gamma^2 \cdot (\sigma \exp(-\lambda)/16)^{k+2}$ with probability at least $1 - \gamma$ over the smoothing of $\mathcal{D}_{\mathbf{x}}$.*

*Proof.* First, from Claim 2.4, it holds that $\mathcal{D}_{\mathbf{x}}$ is $\frac{\exp(-\lambda)}{4}$-unbiased. Since $i \notin \mathsf{supp}(\rho)$, Lemma 2.5 implies that

$$\min(\mathop{\mathbb{E}}_{\mathcal{D}_\rho} [x_i], 1 - \mathop{\mathbb{E}}_{\mathcal{D}_\rho} [x_i]) \geq \frac{\exp(-\lambda)}{4}. \tag{4}$$

Thus, it suffices to lower bound $\mathbb{E}_{\mathcal{D}_\rho}[g_i(x_{-i})]$. Since $x_i$ is relevant, there must exist a restriction $\rho$ with $\mathsf{supp}(\rho) = N_G(i)$ such that the function $(g_i)_\rho$ is not the zero function (otherwise $f(x) = h_i(x_{-i})$ is not dependent on $x_i$). We consider such a restriction $\rho$.

Recall that $f(x) = x_i \cdot g_i(x_{-i}) + h_i(x_{-i})$. Since $f$ is a function on $k$ variables, both $g_i$ and $h_i$ are polynomials such that any non-zero coefficient in both these functions is at least $2^{-k}$ in magnitude. Thus, we have that $|g_i(x)| \neq 0 \implies |g_i(x)| \geq 2^{-k}$.

Let $R$ be the set of relevant variables of $f$. Let $R_i$ be $R \setminus \{i\}$. If $R_i \setminus N_G(i) = \phi$, then it holds that $(g_i)_\rho$ is a constant function with magnitude greater than $2^{-k}$. Thus, combining with Equations (3) and (4), we have that $|\mathbb{E}_{\mathcal{D}_\rho}[y x_i] - \mathbb{E}_{D_\rho}[y] \mathbb{E}_{\mathcal{D}_\rho}[x_i]| \geq 2^{-k} \cdot \frac{\exp(-2\lambda)}{16}$. Thus, it only remains to consider the case where $R_i \setminus N_G(i) \neq \phi$. Let $T_i = R_i \setminus N_G(i)$. Observe that

$$(g_i)_\rho(x) = \sum_{z \in \{0,1\}^{|T_i|}} \mathbb{1}\{x_{T_i} = z\} \cdot h(z) \tag{5}$$

where $h(z) = 0$ or $|h(z)| \geq 2^{-k}$.

Since the marginal $\mathcal{D}_{\mathbf{x}}$ is $(\lambda, \sigma)$-smooth, $\mathcal{D}_{\mathbf{x}}$ has the factorization $\psi(x) = \bar{\psi}(x) + \sum_{i=1}^n \Delta_i x_i$ where $|\partial_i \bar{\psi}(x)| \leq \lambda$ and $\Delta_i = \log(1 + \alpha_i)$ for iid $\alpha_i \sim \mathrm{Unif}([-\sigma, \sigma])$. Let $D_i$ be the MRF with factorization $\psi_i(x) = \psi(x) - \sum_{j \in T_i} \Delta_j x_j$. We have that

$$\mathop{\mathbb{E}}_{\mathcal{D}_\rho} [g_i(x_{-i})] = \sum_{z \in \{0,1\}^{|T_i|}} \mathop{\mathrm{Pr}}_{\mathcal{D}_\rho} [x_{T_i} = z] \cdot h(z)$$

$$= \sum_{z \in \{0,1\}^{|T_i|}} \frac{\mathrm{Pr}_{\mathcal{D}_\rho}[x_{T_i} = z]}{\mathrm{Pr}_{(D_i)_\rho}[x_{T_i} = z]} \cdot \mathop{\mathrm{Pr}}_{(D_i)_\rho} [x_{T_i} = z] \cdot h(z) \tag{6}$$

We now derive an expression for the density ratio. Note that the distribution $(D_i)_\rho$ does not depend on $\Delta_{T_i}$, by definition. Let $Q_i$ denote the set $N_G(i) \cup T_i$. For ease of notation, assume that the elements for $N_G(i)$ come before $T_i$. Let $w_\rho := \rho_{N_G(i)}$ be the string of fixed variables of $\rho$. We have that

$$
\frac{\Pr_{\mathcal{D}_\rho}[x_{T_i} = z]}{\Pr_{(D_i)_\rho}[x_{T_i} = z]} = \frac{\sum_{x_{Q_i}=(w_\rho,z)} \exp(\psi(x))}{\sum_{x_{N_G(i)}=w_\rho} \exp(\psi(x))} \cdot \frac{\sum_{x_{N_G(i)}=w_\rho} \exp(\psi_i(x))}{\sum_{x_{Q_i}=(w_\rho,z)} \exp(\psi_i(x))}
$$

$$
= \exp\Big(\sum_{j \in T_i} \Delta_j z_j\Big) \cdot \frac{\sum_{x_{N_G(i)}=w_\rho} \exp(\psi_i(x))}{\sum_{x_{N_G(i)}=w_\rho} \exp(\psi(x))} \tag{7}
$$

The first equality follows from the expression of the densities of $\mathcal{D}_\mathbf{x}$ and $D_i$. The second inequality follows from the fact that $\psi_i(x) = \psi(x) - \sum_{j \in T_i} \Delta_j x_j$. Note that the second term in the last expression does not depend on $z$. We give a lower bound on this term .

$$
\frac{\sum_{x_{N_G(i)}=w_\rho} \exp(\psi_i(x))}{\sum_{x_{N_G(i)}=w_\rho} \exp(\psi(x))} \geq \min_{x_{N_G(i)}=w_\rho} \exp(\psi_i(x) - \psi(x))
$$

$$
\geq \min_{x_{N_G(i)}=w_\rho} \exp\Big(-\sum_{j \in T_i} \Delta_j x_j\Big) \geq 2^{-k}. \tag{8}
$$

The first inequality follows from the mediant inequality. The second follows from the definition of $\psi_i$ and the last follows from the facts that (1) $\exp(\Delta_j) \leq (1 + \sigma) \leq 2$, and (2) $|T_i| \leq k$.

Now, combining Equations (6) to (8), we obtain that

$$
|\mathop{\mathbb{E}}_{\mathcal{D}_\rho}[g_i(x_{-i})]| = \Big| \sum_{z \in \{0,1\}^{|T_i|}} \exp\Big(\sum_{j \in T_i} \Delta_j z_j\Big) \cdot \frac{\sum_{x_{N_G(i)}=w_\rho} \exp(\psi_i(x))}{\sum_{x_{N_G(i)}=w_\rho} \exp(\psi(x))} \cdot \Pr_{(D_i)_\rho}[x_{T_i} = z] \cdot h(z) \Big|
$$

$$
= \Big| \frac{\sum_{x_{N_G(i)}=w_\rho} \exp(\psi_i(x))}{\sum_{x_{N_G(i)}=w_\rho} \exp(\psi(x))} \Big| \cdot \Big| \sum_{z \in \{0,1\}^{|T_i|}} \exp\Big(\sum_{j \in T_i} \Delta_j z_j\Big) \cdot \Pr_{(D_i)_\rho}[x_{T_i} = z] \cdot h(z) \Big|
$$

$$
\geq 2^{-k} \cdot \Big| \sum_{z \in \{0,1\}^{|T_i|}} \exp\Big(\sum_{j \in T_i} \Delta_j z_j\Big) \cdot \Pr_{(D_i)_\rho}[x_{T_i} = z] \cdot h(z) \Big| \tag{9}
$$

Define $\bar{h}$ to be the function $\bar{h}(z) := \Pr_{(D_i)_\rho}[x_{T_i} = z] \cdot h(z)$. Note that $\bar{h}$ does not depend on $\Delta_{T_i}$ by definition of $D_i$. Combining the fact that $h(z) = 0$ or $|h(z)| \geq 2^{-k}$ and Lemma 2.5, we observe that $\bar{h}(z) = 0$ or $|\bar{h}(z)| \geq 2^{-k} \cdot \frac{\exp(-\lambda|T_i|)}{4^{|T_i|}} \geq \frac{\exp(-\lambda k)}{8^k}$.

Recall that $\Delta_i = \log(1 + \alpha_i)$ for iid $\alpha_i \sim \text{Unif}([-\sigma, \sigma])$. We obtain that with probability at least $1 - \gamma$ over the choice of $\alpha \sim \text{Unif}([-\sigma, \sigma]^n)$, it holds that

$$
|\mathop{\mathbb{E}}_{\mathcal{D}_\rho}[g_i(x_{-i})]| \geq 2^{-k} \cdot \Big| \sum_{z \in \{0,1\}^{|T_i|}} \prod_{j \in T_i} (1 + \alpha_i z_i) \cdot \bar{h}(z) \Big| \geq \gamma^2 \cdot (\exp(-\lambda)\sigma/16)^k
$$

The first inequality follows from the way $\Delta$ is sampled. The second inequality follows from Lemma 3.1 using the facts that (1) the expression on its left hand side is a multilinear polynomial in $\alpha$ of degree at most $k$ with maximal coefficient equal to $|\bar{h}(z)| \geq \frac{\exp(-\lambda k)}{8^k}$ for some $z$, and (2) $\alpha_i$ are sampled iid and uniformly from $[-\sigma, \sigma]$.

Thus, combining everything together, we obtain that for any relevant variable $i$, with probability $1 - \gamma$ over the smooth marginal, there exists a restriction $\rho \in \{0, 1, *\}^n$ such that $\text{supp}(\rho) = N_G(i)$ and it holds that

$$
|\mathop{\mathbb{E}}_{\mathcal{D}_\rho}[yx_i] - \mathop{\mathbb{E}}_{D_\rho}[y] \mathop{\mathbb{E}}_{\mathcal{D}_\rho}[x_i]| \geq \gamma^2 \cdot (\exp(-\lambda)\sigma/16)^{k+2}.
$$

$\square$

## 4 Open Questions

A natural open question is that of learning decision trees over smoothed Markov Random Fields. Recall that Kalai and Teng [KT08] gave a polynomial time learning algorithm for $\log n$-depth decision trees over smooth product distributions. Theorem 3.2 implies polytime learnability for $O(\log \log n)$-decision trees (as every depth $k$ decision tree is a $2^k$-junta), however it is still open how to get all the way to depth $O(\log n)$. Even more ambitiously, one might ask how to learn polynomial size decision trees or poly-size DNFs over smoothed MRFs. Note that there are polynomial time algorithms that learn these classes over smoothed product distributions [KST09].

Finally, an interesting question is if one can avoid the explicit structure learning step in the algorithm. Is there a way to learn juntas over smoothed MRF's without learning the full structure of the distribution?

## Acknowledgments and Disclosure of Funding

Gautam Chandrasekaran is supported by the NSF AI Institute for Foundations of Machine Learning (IFML). Adam Klivans is supported by the NSF AI Institute for Foundations of Machine Learning (IFML).

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

# A Omitted Proofs from Section 2

**Claim A.1.** *Let $\lambda \in \mathbb{R}$. Let $D$ be an MRF with factorization polynomial $\psi$ such that for all $i \in [n]$, it holds that $\|\partial_i \psi\|_1 \leq \lambda$. Then, $D$ is $\frac{\exp(-\lambda)}{2}$-unbiased.*

*Proof.* From, Fact 2.3, we have that for all $i \in [n]$ and $x \in \{0,1\}^{n-1}$, it holds that $\Pr_{X \sim D}[X_i = 1 \mid X_{-i} = x] = \sigma(\partial_i \psi(x)) \geq (1/2) \cdot \exp(-\lambda)$. Similarly, it holds that $\Pr_{X \sim D}[X_i = 0 \mid X_{-i} = x] = 1 - \sigma(\partial_i \psi(x)) \geq (1/2) \cdot \exp(-\lambda)$ □

*Proof of Lemma 2.5.* From the definition of conditional probability, for any $i \in [n]$, set $R \in [n]$, $b \in \{0,1\}$ and $r \in \{0,1\}^{|R|}$, it holds that

$$\Pr_{X \sim D}[X_i = b \mid X_R = r] = \sum_{q \in \{0,1\}^{n-1-|R|}} \Pr_{X \sim D}[X_i = b \mid X_{-i} = (r,q)] \cdot \Pr_{X \sim D}[X_{-i} = (r,q) \mid X_R = r]$$
$$\geq \delta \tag{10}$$

where the first equality follows from expanding the probability and the second inequality follows from the fact that $D$ is $\delta$-unbiased. In the first inequality, we assumed for ease of notation that the elements of $R$ occur before the elements of $[n] \setminus (R \cup \{i\})$.

Without loss of generality, assume that $T = [|T|]$. Then, we have that

$$\Pr_{X \sim D}[X_T = t \mid X_S = s] = \prod_{i \in [|T|]} \Pr_{X \sim D}[X_i = t_i \mid X_{[i-1]} = t_{[i-1]}, X_S = s] \geq \delta^{|T|}$$

where the last inequality follows from Equation (10). □

Finally, we show that the smooth MRFs we consider in this paper are unbiased.

*Proof of Claim 2.4.* Let the factorization of $D$ be $\psi(x) = \bar{\psi}(x) + \sum_{i=1}^{n} \Delta_i x_i$. We have that

$$\Pr_{X \sim D}[X_i = 1 \mid X_{-i} = x] = \sigma(\partial_i \psi(x)) = \frac{\exp(\partial_i \psi(x))}{1 + \exp(\partial_i \psi(x))} \geq \frac{\exp(-|\partial_i \psi(x)|)}{2}$$
$$\geq \frac{\exp(-|\partial_i \bar{\psi}(x)|) \exp(-|\Delta_i|)}{2} \geq \frac{\exp(-\lambda) \cdot \min(\exp(-\Delta_i), \exp(\Delta_i))}{2}$$
$$\geq \frac{\exp(-\lambda)(1 - \sigma)}{2} \geq \frac{\exp(-\lambda)}{4}$$

where we used the definition of $(\sigma, \lambda)$-smooth MRF in the last three inequalities. □

# B Proofs from Section 3.3

We show that with high probability over the smoothing of the distribution, Algorithm 1 run on a sufficiently large number of samples will find all the variables participating in the junta.

**Theorem B.1.** *Let $\mathcal{D}$ be a labelled distribution over $\{0,1\}^n \times \{0,1\}$ such that $\mathcal{D}_{\mathbf{x}}$ is a $(\lambda, \sigma)$-smooth MRF with dependency graph $G$ of degree at most $d$ and the labelling function is a $k$-junta. Then, Algorithm 1 run with $N = \Omega(\text{poly}(\log n, \exp(\lambda(d+k)), 2^{d+k}, \sigma^{-k}, 1/\delta)$ samples from $\mathcal{D}$, graph $G$ and appropriately chosen threshold $\tau$ will find the relevant variables of $f$ with probability at least $1 - \delta$ over the samples and smoothing of $\mathcal{D}$.*

*Proof.* We choose the thresholds $\tau$ such that $2\tau = (\delta/(k2^d))^2 \cdot (\sigma \exp(-\lambda)/16)^{k+2}$. Let $f$ be the $k$-junta generating the labels. Fix a variable $i \in [n]$. We analyze the quantity from Algorithm 1. Recall from Claim 3.4 that with probability $1 - \delta/2$ over the smoothing of $\mathcal{D}_{\mathbf{x}}$, for all coordinates $i$ that are relevant to $f$, there exists a restriction $\rho$ with $\text{supp}(\rho) = N_G(i)$ such that $|I(i, \rho)| \geq 2\tau$. Moreover, from Claim 3.3, we have that $I(i, \rho) = 0$ for any $i$ that is not relevant. Since $|N_G(i)| \leq d$, for all $i \in [n]$, the total number of restrictions enumerated in Algorithm 1 is at most $2^d$ for each $i$. Thus, taking a union bound over all relevant variables and restrictions of their neighbours, Claim 3.4

implies that with probability at least $1 - \delta/2$ over the smoothing of $\mathcal{D}_{\mathbf{x}}$, it holds that for all $i \in [n]$ that are relevant, there exists a restriction $\rho$ with $\mathsf{supp}(\rho) = N_G(i)$ such that

$$|I(i, \rho)| \geq \delta^2 \cdot (\sigma \exp(-\lambda)/16)^{k+2}/(k2^d) = 2\tau. \tag{11}$$

We use concentration of measure to bound the number of samples $N = |S|$ required such that with probability $1 - \delta/2$ over $S \sim \mathcal{D}^{\otimes N}$, it holds that $|I_S(i, \rho) - I(i, \rho)| \leq \tau$ for all $i \in [n]$ and restrictions $\rho$ with $\mathsf{supp}(\rho) = N_G(i)$. We prove the following claim.

**Claim B.2.** *Let $i \in [n]$ and $\rho \in \{0, 1, *\}^n$ with $|\mathsf{supp}(\rho)| \leq d$. Then for $N \geq \Omega((1/\tau^2) \cdot \mathrm{poly}(2^d, \exp(\lambda d)) \cdot \log(1/\gamma))$, it holds that $|I_S(i, \rho) - I(i, \rho)| \leq \tau$ with probability at least $1 - \gamma$ over $S \sim \mathcal{D}^{\otimes N}$.*

*Proof.* Let $T = \mathsf{supp}(\rho)$ and $w_\rho = \rho_T$. From Lemma 2.5, it holds that $\Pr_{X \sim \mathcal{D}_{\mathbf{x}}}[X_T = w_\rho] \geq (\exp(-\lambda)/4)^d$. Thus, for $N \geq \mathrm{poly}(2^d, \exp(\lambda d)) \cdot \log(1/\gamma)$, Hoeffding's inequality implies that with probability $1 - \gamma$ over $S \sim D^{\otimes N}$, we have that $|S_\rho| \geq N \cdot (\exp(-\lambda)/8)^d$. Observe that the distribution of the set $S_\rho$ is identical to a sample from $\mathcal{D}_\rho$ of size $N_\rho = |S_\rho|$. Again, from Hoeffding's inequality, choosing $N_\rho \geq \Omega((1/\tau^2) \cdot \log(1/\gamma))$, it holds that (1) $|\mathbb{E}_{S_\rho}[x_i] - \mathbb{E}_{\mathcal{D}_\rho}[x_i]| \leq \tau/10$, (2) $|\mathbb{E}_{S_\rho}[y] - \mathbb{E}_{\mathcal{D}_\rho}[y]| \leq \tau/10$ and (3) $|\mathbb{E}_{S_\rho}[yx_i] - \mathbb{E}_{\mathcal{D}_\rho}[yx_i]| \leq \tau/10$ with probability $1 - \delta$ over $S_\rho$. Combining these three inequalities, we obtain that $|I_S(i, \rho) - I(i, \rho)| < \tau$ with probability $1 - \delta$ over $S_\rho$. Choosing $N \geq \Omega((1/\tau^2) \cdot \mathrm{poly}(2^d, \exp(\lambda d)) \cdot \log(1/\gamma))$ is sufficient for $N_\rho$ to be large enough with high probability. $\qquad\square$

Setting $\gamma = \delta/(2n2^d)$ in the above claim and taking a union bound over all indices and their corresponding restrictions, we obtain that for $N \geq \Omega(\mathrm{poly}(\log n, \exp(\lambda(d+k)), 2^{d+k}, \sigma^{-k}, 1/\delta))$, with probability at least $1 - \delta$ over the smoothing of $\mathcal{D}_{\mathbf{x}}$ and the sample $S \sim \mathcal{D}^{\otimes N}$, for all $i \in [n]$ and restrictions $\rho$ such that $\mathsf{supp}(\rho) = N_G(i)$, it holds that $|I(i, \rho) - I_S(i, \rho)| < \tau$. In the case of this event, Algorithm 1 succesfully finds all the relevant variables.

$\qquad\square$

Finally, we give the complete proof of the main theorem.

*Proof of Theorem 3.2.* From Theorem B.1, we have that Algorithm 2 run with appropriate parameters succeeds in finding the relevant variables with probability at least $1 - \delta/2$. Now, using a standard concentration arguments (similar to Claim B.2) and taking a union bound over all fixings of the relevant variables, we have that $S$ contains a sample consistent with each assignment of the relevant variables. Thus, the ERM hypothesis $\hat{f}$ will necessarily agree with the true labelling function $f$ on all inputs. Note that computing the ERM is trivial in this case as the subset of the sample containing the different assignments of the relevant variables immediately yields the truth table of the function. $\qquad\square$

