# OpenReview forum: "Learning Juntas under Markov Random Fields"
_NeurIPS.cc/2025/Conference — NeurIPS 2025 poster_

### Official Review · Reviewer_HU3C · 2025-07-01

**Clarity:** 1
**Significance:** 2
**Originality:** 2
**Rating:** 5
**Confidence:** 4

**Summary:**

This paper gives an efficient PAC learner for learning $O(\log n)$-sized juntas when the examples are drawn from a Markov Random Fields. This result extends the setting of learning Boolean concept classes from smoothed product distributions (Kalai and Teng'08).

**Questions:**

1. Line 3, 44: The usage of the term _broad generalization_ in the context of Kalai and Teng (2008) may be misleading. $O(\log n)$-sized juntas are a subclass of size-$n$ decision trees, and the results in this paper do not seem to straightforwardly extend to learning of decision trees. I would invite the authors to at least add a footnote clarifying this issue.

2. Line 20: While learning under different marginal distributions is an important question in its own right - and merits standalone discussion, it is important to note that most prior work focused on learning juntas actually consider the setting of learning under label noise. Curiously, any reference to this entire line of learning and testing juntas is missing from the paper. This should be explicitly addressed in the work (at least with a few notable references), especially considering that the paper is considering an extremely classical problem that has been around in the literature for over three decades.

3. Section 1.1, Lines 30-33: How does $c$ relate to $\Delta$ or $\sigma$? What is $n$ in this context?

4. Line 38: If I were to understand the earlier setup, the joint distribution over the instances is a product distribution. Since we are in the noiseless setting (assuming no label noise), what exactly are we referring to as the marginal distribution in this case?

5. Definition 1.1, Line 51 onwards: The definition needs to be heavily rewritten simply based on the fact that this is the _only_ definition (formal or informal) for undirected graphical models, and the venue of choice is NeurIPS, which is a general purpose machine learning conference, unlike specialized theoretical venues like COLT.

- How do we define a graph $G$ in this context? Are we still considering finite simple graphs? What is the degree $d$ of $G$?
- In the context of this paper, can we consider $G$ to simply be the Boolean hypercube?
- In equation $1$, how exactly is $\psi_S$ defined? In Line 61, the authors note that the linear part of the polynomial $\psi$ is called the external field. What is the _linear part_ of $\psi=\sum_S \psi_S$ in this context? What happens if $\psi_S$ is always non-linear - does that indicate an absence of an external field?

6. Definition 1.2: I believe it needs to be explicitly stated that $D$ is an undirected graphical model.

7. Line 73-74, 79-80: Sources should be cited, or an argument needs to be provided. How is this a generalization of p-biasedness?

8. Line 75, 76: In the absence of normalization constants everywhere, there should be short derivation of the expression for the multiplicative perturbation of the density function. Once again it is ambiguous what perturbing the external field exactly means here.

**Ethical Concerns:**

["NO or VERY MINOR ethics concerns only"]

**Final Justification:**

The authors have addressed all of the concerns I had raised earlier. The technical portion of the paper was already quite strong to begin with, and combined with the earlier clarifications, I believe this is a strong contribution to the field of computational learning theory.

**Limitations:**

It is unclear if the results or techniques in this paper imply anything non-trivial about learning more structured concept classes such as Decision Trees. Can the authors provide experimental evaluations indicating the difficulty of learning juntas in this setting compared to the i.i.d. setting and the smoothed product setting?

**Quality:**

2

**Strengths And Weaknesses:**

# Strengths

The novelty of this paper lies in connecting two important subfields of learning - learning graphical models and learning Boolean concept classes, and providing tools for analysis in this setting.

# Weaknesses

In light of existing results in the literature, the paper does a poor job of motivating why the $(O(1), O(1))$-smoothed MRF setting is important or why Theorem 1.3 is interesting since we are constrained to learning only $O(\log n)$-sized juntas over slightly perturbed MRFs. The paper is also quite poorly presented and there is an extreme lack of clarity in the non-technical sections. Please the questions section for more details.

---

> ### Author Rebuttal · Authors · 2025-07-30
>
> -   Thank you for your detailed review and for highlighting both the novelty and areas for improvement.
> -   "In light of existing results in the literature, the paper does a poor job of motivating why the $(O(1), O(1))$-smoothed MRF setting is important...":
>     -    Could the reviewer please clarify which existing results in the literature they are referring to here?
>     -    We point to various sections of our paper that motivate and highlight the importance of Theorem 1.3. First, the significance of the problem of the junta learning is discussed throughout page 1 of the paper. Prior work on efficient junta learning (in beyond worst case models) have studied only the case of product distributions. Also, the standard benchmark is to learn $O(\log n)$ juntas as even the VC dimension of the class scales with $2^k$. These points are discussed in section 1.1. We highlight the importance of going beyond product distributions in lines 38-41, including the fact that this was explicitly posed as an open question in [KT08]. We believe that MRFs are a natural class of non-product distributions which generalize product distributions. Furthermore $(O(1),O(1))$-MRFs naturally generalize the $O(1)$-smooth product distributions studied in the works of [KT08, KST09, BDM20] (we expand on this point further in a subsequent bullet). Given these reasons, we believe that theorem 1.3 is a significant step in the theoretical study of junta learning.
> -   We agree that there is room for improvement in the clarity of the non-technical sections. We now respond to each of the reviewer's questions:
>     -   "Line 3, 44 ..." : We agree with this point. We had meant a broad generalization on the distributional assumption but now see how this could be misleading. We will add a footnote highlighting that (1) the broad generalization is only on the distrbitional assumoptions and (2) the results don't extend to decision trees.
>     -    "Line 20..." : The focus of our work is on PAC learning (noiseless) juntas. All the difficulties of efficient junta learning are already present in this noiseless case. We also note that as a consequence of our algorithm being in the Statistical query (SQ) framework, it is already noise tolerant to random classification noise [Kea93]. We will add a discussion on this to our paper. Since our focus was on going beyond the $n^k$ runtime (which is a barrier for the noiseless case itself), we had not focussed on label noise variants of the problem. We have discussed the testing problem in Section 1.4 (lines 102-112). We kindly ask the reviewer to point to additional references that they believe are missing in our discussion of related work.
>     -    "Section 1.1, Lines 30-33...": Here, we have made a few typos. It should read "Kalai and Teng [KT08] introduced the notion of a $(c,\sigma)$-smooth product distribution, which is a product distribution with mean vector of the form $\mu=\bar{\mu}+\Delta$ where $\bar{\mu}\in [-c,c]$ is adversarially chosen and $\Delta$ is randomly sampled from the uniform distribution on $[-\sigma,\sigma]$". Here $c$ plays a similar role to $\lambda$ in our definition of $(\lambda,\sigma)$-smoothed MRF. $n$ is the number of input coordinates, as noted in line 11.
>     -    "Line 38": The marginal distribution is the distribution of the input features. Formally in the junta learning problem, we get samples $(X,Y)$ where $X\sim D_{x}$ and $Y=f(X)$ for junta $f$. Here $D_{x}$ is the marginal distribution. This is a distribution on \(\{0,1\}^n\) and was assumed to be product in prior work.
>     - "Definition 1.1, Line 51 onwards":
>        -  Here $G$ is a simple undirected graph. The vertex set is $[n]$ and the graph represents dependencies between different coordinates of the feature vector as highlighted in Definition 1.1. Here $d$ is the max degree of the graph (we missed the word max).
>        -  $G$ is not the boolean hypercube. The vertex set of $G$ is the set $[n]$, corresponding to coordinates of the input vector. The boolean hypercube on the other hand has vertex set \(\{0,1\}^n\).
>        -  As mentioned in lines 57-58, $\psi_S$ are functions that only depend on the coordinates of $x$ in the set $S$. Since $\psi$ is a polynomial of degree atmost $d$, we have $\psi(x)=\sum_{|S|\leq d} c_S\cdot\prod_{i\in S}x_i$ where $c_S$ for some constanst $(c_S)_{|S|\leq d}$.
>       - The linear part of $\psi$ is the polynomial $\sum_{i\in [n]}c_{\{i\}}\cdot x_i$. We will add this to the paper. Yes, if $\psi$ has no linear part, then it's external field is $0$.
>     - "Definition 1.2...": Agreed. We will make this change.
>     -  "Line 73-74, 79-80...": The sources that make this non-degeneracy assumption are listed on line 83. A $p$-biased product distribution is one where $min_{b\in \{\pm 1\}} (Pr[X_i=b])\geq p$ for all $i\in [n]$. The notion of "Unbiased" distribution (Defn 2.2) generalizes this and we prove in Claim 2.4 that low-width MRFs are unbiased with appropriate parameters. We will add a reference to Claim 2.4 (in line 80) to make this clearer.
>     -  "Line 75,76": We are happy to add a short derivation. We hope that our previous comment on the linear part of the polynomial clarified the question about the external field.
>
> - Generalization to Decision trees: Since depth $k$ decision trees are $2^k$- juntas, our result gives an algorithm that runs in time $2^{2^{k}}\cdot \text{poly}(n)$ to learn depth $k$ decision trees. This is polynomial in $n$ as long as $k\leq O(\log \log n)$. This is a weak result as the work of Kalai and Teng [KT09] achieve learnability all the way up to depth $O(\log n)$. Regardless, we will add a discussion on this in our paper.
> - As the focus of our work is theoretical in nature, experimental evidence seems out of the scope of this work. However, we note that lines 192-196 highlight why the proof techniques for product distributions don't translate to our setting.
>
> [KT08] Adam Tauman Kalai and Shang‑Hua Teng. Decision Trees Are PAC‑Learnable from Most Product Distributions: A Smoothed Analysis. arXiv 2008
>
> [KST09] Adam Tauman Kalai, Alex Samorodnitsky and Shang‑Hua Teng. Learning and Smoothed Analysis. FOCS 2009
>
> [BDM20] Alon Brutzkus, Amit Daniely and Eran Malach. ID3 Learns Juntas for Smoothed Product Distributions. COLT 2020
>
> [Kea93] Michael Kearns. Efficient Noise‑Tolerant Learning from Statistical Queries. STOC 1993

---

> > ### Comment · Reviewer_HU3C · 2025-08-02
> >
> > I thank the authors for their detailed clarifications. I believe that with all of the above changes incorporated and points addressed, I no longer have reservations about the paper, and I will raise my score to reflect my opinion.
> >
> > I have one follow-up point:
> >
> > > "Line 38": The marginal distribution is the distribution of the input features.
> >
> > It seems strange to refer to this distribution as a marginal distribution when this is the only distribution being considered. In the agnostic setting where $(X,Y)\sim D$, it makes sense to refer to $D_X$ as the marginal distribution over the input features. I agree that this is a very pedantic matter, but it did make me question whether the setting being considered included label noise or not.

---

> ### Author Response · Authors · 2025-08-05
> **Response to comment**
>
> We thank the reviewer for raising their score. The usage of term "marginal distribution" is quite standard in the PAC learning literature, even in the noiseless case. We agree that it is the only distribution being considered here(and hence may be confusing), however we have used the term to be consistent (and not use a new term just for the noiseless case).

---

### Official Review · Reviewer_PUJX · 2025-07-01

**Clarity:** 2
**Significance:** 3
**Originality:** 3
**Rating:** 4
**Confidence:** 3

**Summary:**

This paper presents a polynomial-time algorithm to learn a "junta" from data generated by an MRF. In this framework, juntas should be understood as classifiers that implement the irrelevance of certain features. The work represents an extension of a previous work roughly corresponding to fully factorised distributions (= MRF with disconnected nodes). This is a technical paper with detailed proofs in the main body, but it completely lacks empirical validation.

**Questions:**

As mentioned, my main issue with the paper is the lack of a clear connection with machine learning and the (consequent?) absence of any experimental analysis. My main question for the authors is whether they considered this link straightforward and decided to keep it implicit, or whether they believe that the result in the paper has an interest by itself from a learning theory perspective, with no need to establish a connection with machine learning.

I would also like to know whether the choice of ending the paper with the statement of the theorem (and its proofs), and without conclusions, reflects the idea that the technical results do not require any additional commentary.

**Ethical Concerns:**

["NO or VERY MINOR ethics concerns only"]

**Final Justification:**

After reading the rebuttals and the other reviews, taking into consideration my low confidence in learning theory, I decide to raise the rating (from 2 to 4) and the significance (from 1 to 3).

**Limitations:**

Yes.

**Paper Formatting Concerns:**

No concerns.

**Quality:**

3

**Strengths And Weaknesses:**

- The extension from the fully factorised case to that of an MRF, under some reasonable assumptions on the resulting joint distribution, is a clear advance in the SOTA.
Yet, the paper seems to focus on the application of the result to machine learning, but there are almost no efforts to highlight the real potential of the result in ML. In this perspective, the lack of any experiment is an evident weakness,  which is, in my view, sufficient to justify a negative recommendation and a low significance (for this particular conference).
Indeed, I could imagine an ML application involving a classification dataset, with the junta playing the role of a feature selection mechanism paired with a classifier. Yet, I doubt whether this nice approach would lead to competitive results: we know from that literature that approaches based on logical formulae are typically sub-optimal in modern ML.

---

> ### Author Rebuttal · Authors · 2025-07-30
>
> - Thank you for your review and for recognizing that extending junta‐learning from product distributions to MRFs is a clear advance in the state of the art.
> - The problem of learning juntas was first introduced in [Blu94, BL97] motivated by the problem of learning in the presence of irrelevant features. We have highlighted this in lines 18-19 of our paper. We will add a concrete example similar to the one suggested by the reviewer to further emphasize this point.
> - Our work is purely theoretical in nature, and given the significance of the junta learning problem (as discussed below) to learning theory, we believe that our results are interesting by themselves, even without empirical validation.
>     - The problem plays a central role in learning theory. Indeed, the influential paper [MOS04] calls the junta learning problem "the single most important
> open question in uniform distribution learning" (see section 1.1 of their paper). We view the extension to non-product distributions as an important next step in our theoretical understanding as most real-world data distributions consist of variables that depend on each other. Our work takes a natural next step, by showing efficient learnability over a wide class of non-product distributions (MRFs).  MRFs are a natural non-product distribution on the boolean hypercube and they have been influential in the field of probabilistic modelling, statistical physics, computer science etc.
> - We believe that the introduction section gives sufficient context and discussion regarding the significance of our main theorem. This is why we state the main theorem in the introduction itself (section 1.3). The succeeding sections give technical details such as the description of the algorithm and supporting proofs.
>
> [Blu94] Avrim Blum. Relevant Examples and Relevant Features: Thoughts from Computational Learning Theory. AAAI Fall Symposium 1994
>
> [BL97] Avrim L. Blum and Pat Langley. Selection of Relevant Features and Examples in Machine Learning. Artificial Intelligence 1997
>
> [MOS04] Elchanan Mossel, Ryan O’Donnell and Rocco A. Servedio. Learning Functions of k Relevant Variables. J. Comput. Syst. Sci. 2004

---

> > ### Comment · Reviewer_PUJX · 2025-08-04
> > **Thanks for the clarification about learning theory, I will change my rating**
> >
> > I thank the reviewers for their comment. We all agree about the technical quality of the paper, and my concern is only about the significance of the topic. The rebuttals and the other reviews already helped me to reconsider my point, especially in light of my low confidence in learning theory. I will raise my rating to a borderline accept. Apart from that, I think I have now a better picture to evaluate the work of the authors and discuss it with the other reviewers. I don't' think I need additional discussion with the authors.

---

### Official Review · Reviewer_RNxT · 2025-07-02

**Clarity:** 3
**Significance:** 3
**Originality:** 2
**Rating:** 5
**Confidence:** 3

**Summary:**

This paper investigates the following problem: Given random samples i.i.d. drawn from a Markov Random Field (with a known dependency graph of degree $O(\log n)$), under a smoothed analysis framework, and labeled by an unknown k-junta, is there an efficient algorithm to recover the underlying k-junta? This work represents a significant generalization of prior research, which was largely confined to product distributions. A key contribution is the introduction of a conditional correlation statistic, $I(i,\rho)$, which effectively leverages the property of MRFs to distinguish relevant features from irrelevant ones. Building on this statistic, the authors develop an algorithm with both runtime and sample complexity bounded by $n^{o(k)}$.

**Questions:**

The method is designed specifically for learning k-juntas, which is a relatively restricted model. It would be helpful if the authors could clarify what makes it particularly challenging to recover more complex concept classes, such as k-depth decision trees or k-DNFs under MRF distributions.

In line 162, the lowercase variables  s and t appear to be undefined in the surrounding context, this seems to be a typographical error.

**Ethical Concerns:**

["NO or VERY MINOR ethics concerns only"]

**Final Justification:**

The generalization from prodcut distribution to any given MRF is natural, and the technical approach is also interesting. The authors also does a decent job of literature review. I recommend acceptance for this paper.

**Limitations:**

yes

**Paper Formatting Concerns:**

Nothing major

**Quality:**

3

**Strengths And Weaknesses:**

Strengths:
A nice and natural generalization of product distribution in terms of MRF beyond worst-case analysis

Weaknesses:
As the authors themselves acknowledge, a notable limitation of the proposed method is its reliance on the assumption that the MRF’s dependency graph is known. In the absence of this knowledge, one would need to resort to structure learning techniques for bounded-degree MRFs, incurring substantial additional computational and sample complexity.

---

> ### Author Rebuttal · Authors · 2025-07-30
>
> -    Thank you for your review and positive feedback.
> -    We acknowledge the weakness regarding the substantial additional cost if the underlying graph is not known. It is indeed an interesting open problem if structure learning is necessary for this task.
> -    Difficulty in extending to decision trees/DNF:  One main problem is that the number of relevant variables can now be $2^{k}$ as opposed to $k$. Thus, applying our method naively will incur a run time of $2^{2^{k}}$. Kalai and Teng [KT08] relied heavily on Fourier techniques to learn decision trees over smooth product distributions. Their arguments seem to rely crucially on the orthogonality of the fourier basis. These Fourier arguments don't extend to MRFs as there is no explicit orthogonal basis. It is an interesting open question if we learn decision trees (even in the case of smooth product distributions) without the use of Fourier techniques. We will add a discussion on this to the paper.
>
> [KT08] Adam Tauman Kalai and Shang‑Hua Teng. Decision Trees Are PAC‑Learnable from Most Product Distributions: A Smoothed Analysis. arXiv 2008

---

> > ### Comment · Reviewer_RNxT · 2025-08-05
> >
> > Thanks for the response, I have read the rebuttal and do not have additional concerns. I will keep my positive rating and recommendation.

---

### Official Review · Reviewer_Spcm · 2025-07-03

**Clarity:** 3
**Significance:** 2
**Originality:** 3
**Rating:** 4
**Confidence:** 3

**Summary:**

This paper studies a theoretically deep question: the $k$-junta problem where marginal distributions are perturbed instead of uniform over the hypercube; in particular, the problem in this paper is over Markov random fields (MRF). The main result yields a successful two-phased algorithm (with high probability) with sample size $\Omega(\text{poly}(\log n, \exp(\lambda(d+k)), 2^{d+k}, \sigma^{-k}, 1/\delta))$. Their result is a natural generalization of earlier works for smoothed product distributions.

**Questions:**

1. Please see Weakness
2. Does the learning of $k$-junta problem have any relevance with the task of learning $k$-parity, or do results for one problem inspire the other? If the answer is yes, then discussions for $k$-parity learning may be helpful.

**Ethical Concerns:**

["NO or VERY MINOR ethics concerns only"]

**Final Justification:**

The authors addressed my concern and I have decided to keep my score.

**Limitations:**

Yes

**Quality:**

3

**Strengths And Weaknesses:**

Strengths:
1. The generalization is natural and significant: Extending the problem scale to a broader class of distributions is a long standing open problem, and the authors gave an elegant solution and sound analysis.
2. The techniques illustrated in Section 3 is theoretically rich and novel. It is refreshing by blending unsupervised and supervised methods. The algorithm, despite requiring a possibly high sample size (see Weakness), will very likely stimulate further research on this area and further optimization.

Weakness:
My major concern is the size of $k$; it is only logarithmic and therefore may restrict the paper's scope. I understand that this constraint is necessary to bound sample complexity to polynomial, but, to the best of my knowledge, problems in similar themes ($k$-parity, $k$-junta, etc) usually assume $k$ to be a constant fraction of $n$.

---

> ### Author Rebuttal · Authors · 2025-07-30
>
> - Thank you for your review and encouraging feedback.
> -   Concern about size of $k$: The regime $k\leq O(\log n)$ is the main regime of parameters studied in the literature of efficient junta learning algorithms. As correctly observed by the reviewer, going beyond this regime will require superpolynomial sample complexity for generalization. Thus, even information theoretically, this is the only regime where one can learn with polynomial samples. We are not aware of the line of work that the reviewer refers to, where $k$ is assumed to be a constant fraction of $n$.
> -   Yes, learning $k$-junta does have relevance to the $k$-parity problem. In fact any SQ algorithm for learning $k$-juntas also works for learning noisy $k$-parities (as $k$-parities are $k$-juntas and SQ algorithms are tolerant to random classification noise). Since our algorithm falls in the SQ framework, it works even for the noisy $k$-parity problem. We will add a discussion on this.

---

> > ### Comment · Reviewer_Spcm · 2025-08-06
> > **Reply to rebuttal**
> >
> > I thank the authors for clarifying my concerns. I’m happy to keep my score

---

### Decision · Program_Chairs · 2025-09-17

**Decision:**

Accept (poster)

**Comment:**

This paper examines the problem of efficiently learning k-juntas in polynomial time when the marginal distribution follows a discrete Markov Random Field (MRF). This is a classical problem where efficient algorithms, beyond the naive exhaustive search, had only been established for the limited case of product distributions. The authors bridge this gap by presenting an efficient algorithm that extends to the broader class of MRF-type distributions, although the result depends on a specific perturbation method.

Four reviews were collected for this submission. All reviewers highlighted the technical depth of the work and the novelty of the proposed theoretical framework. The AC concurs, viewing this as a valuable theoretical contribution that introduces several new ideas. While reviewers raised some concerns and follow-up questions (for example, regarding the significance of the results and clarity in certain sections), the authors addressed these thoroughly in their response.

The AC thanks the authors for their strong contribution.